# A New Method for Conditional Gene-Based Analysis Effectively Accounts for the Regional Polygenic Background

**DOI:** 10.3390/genes15091174

**Published:** 2024-09-07

**Authors:** Gulnara R. Svishcheva, Nadezhda M. Belonogova, Anatoly V. Kirichenko, Yakov A. Tsepilov, Tatiana I. Axenovich

**Affiliations:** 1Institute of Cytology and Genetics, Siberian Branch of Russian Academy of Sciences, Ave. Lavrentiev, 10, 630090 Novosibirsk, Russia; belon@bionet.nsc.ru (N.M.B.); kianvl@bionet.nsc.ru (A.V.K.); tsepilov@bionet.nsc.ru (Y.A.T.); aks@bionet.nsc.ru (T.I.A.); 2Institute of General Genetics, Russian Academy of Sciences, Gubkin St. 3, 119311 Moscow, Russia; 3Wellcome Sanger Institute, Wellcome Trust Genome Campus, Cambridge CB10 1RQ, UK

**Keywords:** random-effects model, conditional distribution, gene-based association analysis, summary statistics

## Abstract

Gene-based association analysis is a powerful tool for identifying genes that explain trait variability. An essential step of this analysis is a conditional analysis. It aims to eliminate the influence of SNPs outside the gene, which are in linkage disequilibrium with intragenic SNPs. The popular conditional analysis method, GCTA-COJO, accounts for the influence of several top independently associated SNPs outside the gene, correcting the z statistics for intragenic SNPs. We suggest a new TauCOR method for conditional gene-based analysis using summary statistics. This method accounts the influence of the full regional polygenic background, correcting the genotype correlations between intragenic SNPs. As a result, the distribution of z statistics for intragenic SNPs becomes conditionally independent of distribution for extragenic SNPs. TauCOR is compatible with any gene-based association test. TauCOR was tested on summary statistics simulated under different scenarios and on real summary statistics for a ‘gold standard’ gene list from the Open Targets Genetics project. TauCOR proved to be effective in all modelling scenarios and on real data. The TauCOR’s strategy showed comparable sensitivity and higher specificity and accuracy than GCTA-COJO on both simulated and real data. The method can be successfully used to improve the effectiveness of gene-based association analyses.

## 1. Introduction

Gene-based association (GBA) analysis is widely used for gene mapping. The interpretation of its results essentially relies on reducing the influence of extragenic SNPs that are in linkage disequilibrium (LD) with internal SNPs of a gene. This is achieved by conditional analysis. GBA analysis is increasingly being performed using GWAS summary statistics and correlation matrices (also called LD matrices) between SNP genotypes. This approach has many advantages over the analysis of individual data [1,2]. A solution to the problem of conditional analysis using GWAS summary statistics was first proposed in [3], where the GCTA-COJO (or COJO for short) method was introduced.

The essence of COJO is to adjust the summary statistics of intragenic SNPs to ensure their independence from the effects of extragenic SNPs. To do this, COJO selects independently associated SNPs from the region surrounding a gene of interest. Then, for each SNP within the gene, COJO recalculates the summary statistics conditional on a given list of top SNPs outside the gene. These conditional summary statistics, along with the original LD matrices, are then used as the input for secondary GBA analysis that can be made by any GBA test. To compute the conditional summary statistics, COJO relies on a multiple linear regression fixed effects model. This model is known for its shortcomings, most notably collinearity and sensitivity to outliers. COJO attempts to address these problems by filtering out the highly correlated extragenic SNPs using a forward stepwise model selection procedure. However, this procedure is classified as an overly greedy algorithm [4]. This means that COJO may miss some potentially helpful SNPs due to their LD with previously detected SNPs [5]. Moreover, COJO is sensitive to the parameters of the model embedded in COJO, resulting in an unstable list of top SNPs. Consequently, there is a risk of overfitting, especially if too many predictors are included in the model.

COJO’s alternative for conditional GBA analysis is the polygene pruning (PP) method described in [6,7]. This method, like COJO, uses summary statistics and is compatible with all GBA tests. The essence of PP is to exclude the intragenic SNPs that are in high LD with more significant SNPs outside the gene. Unlike COJO, which adjusts the summary statistics of intragenic SNPs, PP filters SNPs within a gene, leaving SNPs that are statistically independent of SNPs outside the gene. PP is fast because it does not require complex matrix manipulations. This feature is advantageous when analyzing dense genomic regions containing a large number of SNPs, as the number of predictors in the regression analysis can be significantly reduced after PP. In addition, PP does not require strict inconsistency between summary statistics and reference LD matrices. However, excluding SNPs always reduces the informativity of data sets and might lead to a loss of statistical power compared to methods that focus on correcting summary statistics.

Another method for conditional GBA analysis using summary statistics has been proposed by [8]. This method, however, can be applied only to a particular GBA test, the effective chi-squared statistic (ECS), proposed therein. It therefore precludes the application of all known popular GBA tests, including the Burden test [9], SKAT [10,11], SKAT-O [12,13], PCA [14], FLM [15], and others.

In this paper, we propose another method for conditional GBA analysis using summary statistics. The new method, named TauCOR, aims to account for the external polygenic background in the LD matrix for intragenic SNPs. Unlike COJO, which corrects the GWAS summary statistics for each SNP individually, the new method corrects the whole LD matrix that is attributed to the gene. This matrix is used along with the initial GWAS summary statistics for further secondary GBA analysis. TauCOR is compatible with any linear regression-based GBA test that uses summary statistics and LD matrices as input, and thereby is universal.

The performance of the new method in comparison with COJO was evaluated on the simulated summary statistics and on causal genes from the ‘gold standard’ causal gene list and non-causal genes from neighboring regions.

## 2. Materials and Methods

### 2.1. The TauCOR Method

We focused on two objects: a gene and its surrounding region. SNPs within the gene will be referred to as intragenic or internal, while SNPs from the region surrounding the gene will be referred to as extragenic or external.

#### 2.1.1. Algorithm

For each gene, the new method employs an algorithm comprising two steps (see Figure 1):(i)Estimating the joint contribution of extragenic SNPs to the trait variation and calculating the trait variance explained by these extragenic SNPs (i.e., the local SNP heritability, in terms proposed by Shi et al. [16]), and(ii)Adjusting the LD matrix for intragenic SNPs so that the distribution of the z statistics of these SNPs becomes conditionally independent of the distribution of the z statistics of SNPs in the external region.

The first step of TauCOR relies on the variance-component (VC)-based model, which is a linear regression model with random effects and describes the joint distribution of the z statistics of extragenic SNPs. VC-based tests are widely used in association analysis due to their robust statistical power, even when the region under analysis has many non-causal SNPs and/or when the causal SNPs have different directions and different magnitudes of association [17,18]. In the context of the VC-based model, the parameter of interest is a scalar, *τ*, which reflects the local SNP heritability. The second step also relies on the VC-based model. In this case, however, we are dealing with a conditional model that describes the joint distribution of z statistics of intragenic SNPs, conditioned on the regional polygenic background.

#### 2.1.2. Task: Designations, Input Data, and Formulation

Let us consider a set of *m_g_* SNPs within a gene (denoted by setG) and a set of *m_r_* SNPs from the region around the gene (denoted by setR), *m = m_r_* + *m_g_*. We denote the vectors of SNP-level z statistics as *z_r_* for setR and *z_g_* for setG. We signify the matrices of SNP-SNP correlations within the gene as *U_g_*, within the region around the gene as *U_r_*, and between the gene and the region as *U_rg_*. Here, we distinguish between the three types of z statistic distributions with respect to setG and setR: marginal, conditional, and joint. For the sake of convenience, they are symbolically denoted as *f*(*z_g_*) or *f*(*z_r_*), *f*(*z_g_*|*z_r_*), and *f*(*z_g_*, *z_r_*), respectively. In terms of these designations, our objective is to estimate the conditional distribution *f*(*z_g_*|*z_r_*).

For building the heritability model, we consider a sample of *n* unrelated individuals with measured trait values, *y*, and measured genotypes for setR, *G_r_*, and for setG, *G_g_*. To describe the joint influence of setG and setR on the trait, we employ a linear regression model, in which we assume that the *G_r_* effects are random, while *G_g_* effects can be either random or fixed. This model uses a VC approach and allows us to consider the effects of extragenic SNPs as the external polygenic background that can distort the LD matrix for intragenic SNPs.

For standardized individual data, the model is of the following form:(1)$$\bar{y}=\frac{1}{\sqrt{n}}{\bar{G}}_{g}{\bar{\beta }}_{g}+\frac{1}{\sqrt{n}}{\bar{G}}_{r}{\bar{\beta }}_{r}+\xi_{n}.$$

Here, $\bar{y}$ is an (*n* × 1) vector of standardized trait values at *n* individuals; ${\bar{G}}_{g}$ (or ${\bar{G}}_{r}$) is an (*n × m_g_*) (or (*n × m_r_*)) matrix of standardized genotypes for setR (or setG); ${\bar{\beta }}_{r}$ is an (*m_r_* × 1) vector of random effects of SNPs from setR, ${\bar{\beta }}_{r}\sim N(0,{\tau I}_{m_{r}})$, where *I_k_* is an (*k* × *k*) identity matrix, and *τ* measures a common contribution of SNPs from setR to the trait variability; and ${\bar{\beta }}_{g}$ is an (*m_g_* × 1) vector of random or fixed (depending on the selected GBA test) effects of SNPs from setG. It is important to note that using standardized individual data leads to standardized values of ${\bar{\beta }}_{r}$ and ${\bar{\beta }}_{g}$. Furthermore, for ease of interpretation, we scaled them by $1/\sqrt{n}$ to be able to express them in terms of the unstandardized (original) effect sizes and their standard error (*se*) namely as ${\bar{\beta }}_{r}=\frac{\beta_{r}}{se\left(\beta_{r}\right)}$ and ${\bar{\beta }}_{g}=\frac{\beta_{g}}{se\left(\beta_{g}\right)}$. By definition, ${\bar{\beta }}_{r}$ and ${\bar{\beta }}_{r}$ are equivalent to the joint z statistics. Finally, *ξ_n_* is an (*n* × 1) vector of random standardized regression residuals, $\xi_{n}\sim N\left(0,I_{n}\right)$.

In accordance with [1], Model (1) can be reformulated in terms of summary-level data (for details, see Appendix B) divided into blocks linked with setG or setR:(2)$$\begin{array}{c} setG\to \;\;\\ setR\to \;\; \end{array}\;\;\;\;\;\left(\begin{array}{c} z_{g}\\ z_{r} \end{array}\right)=\underset{\begin{array}{c} ↑\\ setG \end{array}}{\underbrace{\left(\begin{array}{c} U_{g}\\ U_{rg} \end{array}\right)\;{\bar{\beta }}_{g}}}\;\;\;\;+\;\;\;\;\underset{\begin{array}{c} ↑\\ setR \end{array}}{\underbrace{\left(\begin{array}{c} U_{gr}\\ U_{r} \end{array}\right)\;{\bar{\beta }}_{r}}}\;+\left(\begin{array}{c} \xi_{m_{g}}\\ \xi_{m_{r}} \end{array}\right)\;\;\;\;\;\;\;\;\;\;\;$$

Here, $\left(\begin{array}{c} \xi_{m_{g}}\\ \xi_{m_{r}} \end{array}\right)$ is an (*m* × 1) vector of random regression residuals distributed as $N\left(\left(\begin{array}{c} 0\\ 0 \end{array}\right),\left(\begin{array}{cc} U_{g} & U_{gr}\\ U_{rg} & U_{r} \end{array}\right)\right)$.

Model (2) describes the joint distribution *f*(*z_g_*, *z_r_*). In order to construct the conditional distribution *f*(*z_g_*| *z_r_*), we estimate the marginal distribution *f*(*z_r_*) on the basis of Model (2). For this, we focus the bottom row of Expression (2), which corresponds to setR, under the null GBA analysis hypothesis (${\bar{\beta }}_{g}$ = 0):(3)$$z_{r}=U_{r}{\bar{\beta }}_{r}+\xi_{m_{r}}.\;$$

Here, ${\bar{\beta }}_{r}$ is distributed as described above in Model (1), ${\bar{\beta }}_{r}\sim N(0,I_{m_{r}})$, where *τ* is an unknown model parameter that is proportional to the local SNP heritability, $h_{r}^{2}$, explained by setR, and $\xi_{m_{r}}$ is an (*m_r_ ×* 1) vector of random regression residuals, $\varepsilon_{m_{r}}$~ *N*(**0**, *U_r_*). It can be shown that $\tau =\frac{nh_{r}^{2}}{m_{r}}$ (for details, refer to Appendix C). Consequently, certain limitations are placed on the estimation of *τ*, $0\leq \tau \leq \frac{n}{m_{r}}.$

In accordance with Model (3), the marginal distribution of *z_r_* is described by distribution parameters:(4)$$f\left(z_{r}\right)⇐\left\{\begin{array}{c} E(z_{r})=0,\\ E(z_{r}z_{r}^{T})=\tau U_{r}U_{r}+U_{r} \end{array}\right.$$

Here, the symbol ⇐ indicates that the distribution has a given mean vector *E*(*z_r_*) and covariance matrix *E*(*z_r_z_r_*^T^). The scalar *τ* can be estimated numerically from *f*(*z_r_*) described in Expression (4) using the maximum-likelihood estimator (MLE):(5)$$-2\ln Lh\;\sim \;log\left|\tau U_{r}U_{r}+U_{r}\right|+z_{r}^{T}{\left(\tau U_{r}U_{r}+U_{r}\right)}^{-1}\;z_{r}.\;$$

As can be seen, Expression (5) includes a matrix inversion procedure that can be complicated due to multicollinearity of genotypic data. To avoid this problem, we compute a pseudo-inverse matrix for the low-rank approximation obtained from the original matrix (see Appendix A for details).

We then derive a conditional distribution *f*(*z_g_|z_r_*) from the known joint distribution *f*(*z_g_,z_r_*), given the known marginal distribution *f*(*z_r_*). In order to achieve this, we consider the upper row of Model (2), which relates to *z_g_*, and demonstrates three potential sources of trait variation:(6)$$z_{g}=\underset{gene}{\underbrace{U_{g}{\bar{\beta }}_{g}}}+\;\underset{region}{\underbrace{U_{gr}{\bar{\beta }}_{r}}}\;+\underset{others}{\underbrace{\xi_{m_{g}}},}$$

Here, the distribution of ${\bar{\beta }}_{r}$ is already defined by the estimated *τ* parameter as described in Model (1), and $\xi_{m_{g}}$ represents a vector of random regression residuals caused by non-genetic or genetic, but not setG- and setR-associated, factors, $\xi_{m_{g}}\sim \;N\left(0,\;U_{g}\right)$.

As follows from Model (6) under the null hypothesis of GBA analysis (${\bar{\beta }}_{g}$ = 0), the covariance matrix for setG is expressed as follows:(7)$$E\left(z_{g}z_{g}^{T}\right)=\tau U_{gr}U_{gr}^{T}+U_{g.}$$

Then, the conditional distribution *f*(*z_g_|z_r_*) under ${\bar{\beta }}_{g}$ = 0 is given by the distribution parameters:(8)$$f\left(z_{g}|z_{r}\right)⇐\left\{\begin{array}{c} E\left(z_{g}|z_{r}\right)=0,\\ Cov\left(z_{g}|z_{r}\right)=\tau {\;U}_{gr}U_{gr}^{T}+U_{g}.\; \end{array}\right.$$

The next step is to use the initial marginal z statistics, $z_{g}$, and the adjusted $\left(\tau {\;U}_{gr}U_{gr}^{T}+U_{g}\right)$ matrix instead of initial $U_{g}$ matrix as the input for secondary GBA analysis (see Figure 1). This analysis can be performed with any of the GBA tests that use summary data. The most popular GBA tests have been implemented in the sumFREGAT R-package (version 1.2.5) [2].

TauCOR has a property that depends on the GBA test selected. When kernel-based score tests, such as Burden, SKAT, or SKAT-O, are used in conditional GBA analysis, the *p*-values are guaranteed to be greater than or equal to the *p*-values of the initial GBA analysis (a derivation is provided in Appendix A). However, TauCOR loses this property when the PCA test is used (see Appendix A).

### 2.2. Simulation Strategy

We constructed a causal SNP model for all SNPs of a gene and the surrounding region. This model simulates three vector variables: (i) the causal status of SNPs labelled as *c*; (ii) joint z statistics, and (iii) marginal z statistics.

Consider a gene with *m_g_* SNPs and the surrounding region with *m_r_* SNPs. The scheme of distribution of causal SNPs in the gene and surrounding region was as follows:(9)$$\underset{all\;SNPs}{\underbrace{m_{r}+m_{g}}}\to \left\{\begin{array}{c} \underset{causal\;}{\underbrace{K}}\;\;\;\;\;\;\;\to \;\left\{\begin{array}{c} \underset{\;\;in\;gene}{\underbrace{\rho K}}\\ \underset{in\;region}{\underbrace{\left(1-\rho \right)K}} \end{array}\right.\\ \underset{non-causal}{\underbrace{\left(m_{r}+m_{g}\right)-K}}\to \;\left\{\begin{array}{c} \underset{\;\;in\;gene}{\underbrace{m_{g}-\rho K}}\\ \underset{\;\;in\;region}{\underbrace{m_{r}-\left(1-\rho \right)K}} \end{array}\right. \end{array}\right.$$

We describe Scheme (9) using two parameters, *K* and *ρ*. The parameter *ρ*, which varies between 0 and 1, is employed to indicate the location of causal SNPs. The value of *ρ* is 1 if the causal SNPs are located inside the gene, and 0 if the SNPs are located outside the gene. The value of *K* is the total number of causal SNPs. Consequently, *ρK* is the number of causal SNPs in the gene.

The causal statuses of the SNPs in the gene and the surrounding region were modelled separately using a Bernoulli distribution:$$c=\left\{\begin{array}{c} Ber\;\left(\rho K/m_{g}\right),\;\mathrm{if}\;i\in \mathrm{setG}\\ Ber\;\left(\left(1-\rho \right)K/m_{r}\right),\;\mathrm{if}\;i\in \mathrm{setR} \end{array}.\right.$$

In our study, we propose that the heritability explained by a single SNP, $h_{SNP}^{2}$, is the same for all SNPs. This implies that the joint contribution of SNPs from setR to trait variability, *τ*, can be defined via genome-wide heritability, $h_{GW}^{2}$, as follows:(10)$$\tau =\frac{n}{K}\left(\frac{m_{r}+m_{g}}{M}h_{GW}^{2}\right).$$

The causal statuses of the SNPs in the gene and the surrounding region were modelled separately using a Bernoulli distribution, where *M* is the total number of SNPs in the genome, and $\frac{m_{r}+m_{g}}{M}h_{GW}^{2}$ is the local heritability explained by *m_r_ + m_g_* SNPs. For all causal SNPs, we simulated joint z statistics (denoted as *z_j_*) which, by definition, are equal to $\beta /se\left(\beta \;\right)$:$$z_{j}\;\sim \left\{\begin{array}{c} 0,{if\;c}_{i}=0\\ N\left(0,\tau \right),\;if\;c_{i}=1 \end{array}\right.$$

Next, for all SNPs, we modelled the marginal z statistics as *z* ~ *N*(*Uz_j_*, *U*), where *U* is the LD matrix common for both intragenic and extragenic SNPs. These z statistics are equivalent to the z statistics calculated in GWAS. They can be used for GBA analysis.

In our study, we directly simulated summary statistics using real LD matrices for genes and their surrounding regions, which were calculated using genotypes from the 1000 Genomes Project [19] and PLINK (version 1.9) [20]. The direct simulation of marginal z statistics is a valid approach because it has been analytically proven that the distribution of marginal z statistics, expressed via summary statistics, $z\;\sim N\left(US^{-1}\beta ,U\right)$ where *S* is a diagonal matrix with diagonal elements equal to *se*(β) and $S^{-1}\beta =z_{j}$, is identical to the distribution calculated from individual phenotypic values [1]. The marginal SNP effects were calculated as *Sz*.

The value of *K* was fixed at 10. Two classes of scenarios were considered with respect to the *ρ* parameter, which was set to either 0 or 1. Formula (10) was used to assign *τ*, with *h_GW_*^2^ set at 0.3, 0.5, or 0.7. A more detailed description of the parameters and inputs for the simulation is provided in Appendix A. Three GBA tests were selected for initial and conditional GBA analyses, each employing a distinct strategy for detecting association signals: the principal component analysis test (PCA), the Burden test (BT), and the sequence kernel association test (SKAT) implicated in the sumFREGAT package (version 1.2.5) [2,6]. A total of six scenarios for each GBA test were therefore defined by two parameters, *ρ* and *h_GW_*^2^.

For our simulations, we considered real genes on chromosome 22 and surrounding regions of ±1 Mb in size. We selected only genes with the number of internal SNPs varying from 100 to 500 and with the number of adjacent external SNPs varying from 4000 to 9000. The total number of such genes was 54. The input data for the simulation analysis of a single gene were the LD matrices for setR and setG, *U_r_* and *U_g_*, respectively: the LD matrix between setR and setG (*U_rg_*); the ratio of the sample size to the total number of SNPs (n/M = 0.05); and the simulation model parameters (*ρ*, *h_GW_*^2^ and *K*).

A total of 2000 runs were conducted to simulate association signals in a gene (*ρ* = 1), and 6000 runs were conducted in the region surrounding this gene (*ρ* = 0), for each of the 54 genes. For a conditional analysis, only those runs were selected in which the gene exhibited a significant association signal (*p*-value < 2.5 × 10^−6^).

For the COJO-based analysis of a gene, the surrounding region was initially examined to determine any conditional SNPs using the ‘--cojo-slct’ option. The *p*-value threshold given by the ‘--cojo-p’ option was set to 1.0 × 10^−4^. If no conditional extragenic SNPs were detected, the gene was considered as having passed the COJO-based analysis, with its initial gene-based *p*-value remaining unchanged. If the number of conditional extragenic SNPs after ‘--cojo-slct’ exceeded 10, the 10 strongest were selected. With the final list of conditional extragenic SNPs and the ‘--cojo-cond’ option, the corrected summary statistics for the intragenic SNPs were obtained and used as the input for subsequent secondary GBA analysis. If the recalculated GBA *p*-value was statistically significant (≤2.5 × 10^–6^), the gene was considered as having passed the COJO-based analysis.

For the TauCOR-based analysis, the correlation matrix between intragenic SNPs and extragenic SNPs within a given window was used to estimate *τ* and calculate the corrected correlation matrix for SNPs within the gene. Together with the original z statistics, this corrected correlation matrix was used as the input to the secondary GBA analysis. If the GBA test *p*-value was statistically significant (≤2.5 × 10^–6^), the gene was considered as having passed the TauCOR-based analysis.

### 2.3. ‘Gold Standard’ Gene List

In addition to the simulation data, we employed real data to assess the characteristics of the novel method.

We selected a list of 28 ‘gold standard’ (GS) genes from the Open Targets Genetics project [21], i.e., genes whose causal effect on a trait is clearly established. They were considered to be causal genes in this study. These genes were associated with 13 traits. The GWAS for these traits were selected from the UK Biobank (https://pheweb.org/UKB-SAIGE/, accessed on 31 May 2023). Genes sampled in 1 Mb regions around GS genes were considered non-causal. A total of 394 genes with more than one SNP were identified within all regions in addition to the GS genes. The number of such genes per region ranged from one to 59, with an average of 14.6.

For all selected genes, we performed GBA analysis using the sumSTAAR framework [6]. For each gene, SNPs were filtered by MAF ≤ 10^−4^, annotated using the VEP tool (version 107) [22], and divided into three categories (sets) of SNPs: non-coding, synonymous, and non-synonymous variants.

GBA analysis was carried out using the ACAT-O combination of six tests: SKAT-O (optimal combination of BT and SKAT) and PCA testing for three SNP sets. Genes that reached the standard GBA significance threshold (*p* < 2.5 × 10^−6^) were selected for conditional analyses. Two methods for conditional analysis were used: COJO, as implemented in the GCTA tool (version 1.25.0), and TauCOR. For each gene, a window with 5 Mb or 1.5 Mb from both gene boundaries was used for COJO or TauCOR, respectively.

Conditional COJO and TauCOR analyses were performed as described in the previous section, except the threshold *p*-value for COJO selection of extragenic SNPs that was defined as the minimum *p*-value among intragenic SNPs.

### 2.4. Method Performance

A generally accepted significance threshold of 2.5 × 10^−6^ was used to determine positive (i.e., trait-associated) (*p* < 2.5 × 10^−6^) and negative (i.e., non-associated) genes (*p* ≥ 2.5 × 10^−6^). The sensitivity of the analysis was evaluated on the set of genes in which the effect was simulated (*ρ* = 1) and on the set of GS genes. The sensitivity was calculated as the ratio of the number of associated genes to the total number of genes involved in the analysis. Specificity was evaluated on the set of genes in which the signal was not simulated (when *ρ* = 0) and the set of genes surrounding the GS genes, as the ratio of the number of non-associated genes to the total number of genes involved in the analysis. Finally, accuracy was calculated as the proportion of genes in which the analysis result matched the expected result.

## 3. Results

### 3.1. Simulated Data Analysis

Figure 2 presents the results of the simulation analysis, which include estimates of sensitivity, specificity, and accuracy for two conditional GBA analysis methods under different scenarios. Further details on the simulation results can be found in Appendix A. These tables include the number of gene runs selected for conditional analysis (*N_an_*), as well as estimates of performance measures for an initial GBA analysis and two conditional GBA analyses under different scenarios.

Initial GBA analysis. The initial GBA tests showed consistently high specificity (>90%) across all scenarios. However, the sensitivity of the different tests varied. As anticipated, SKAT and PCA tests that support the bidirectionality of the causal SNP effects showed moderate to high sensitivity, ranging from 61% to 91%. In contrast, BT demonstrated lower sensitivity, ranging from 26% to 44%. Nevertheless, the accuracy, calculated essentially as a linear combination of sensitivity and specificity, was acceptable, exceeding 85% for all tests. Moreover, as expected, for each of the GBA tests, there was a decrease in specificity and an increase in sensitivity with increasing *h_GW_*^2^ (Appendix A).

Conditional GBA analysis. COJO sometimes failed to process runs with initial GBA *p*-values below 1.0 × 10^−30^; therefore, only runs with an initial GBA *p*-value below 2.5 × 10^−6^ but above 1.0 × 10^−30^ were permitted for conditional analysis.

For PCA and SKAT testing, TauCOR sensitivity was high and comparable to COJO sensitivity (>93%), whereas for BT, TauCOR sensitivity was significantly higher than COJO sensitivity. Overall, TauCOR showed an acceptable sensitivity of over 86% in all scenarios. However, the specificities of COJO and TauCOR did not exceed 60% in all scenarios. In particular, with regard to BT, the specificities of COJO and TauCOR were found to be similar, with approximately 50% observed for both. In contrast, for PCA testing, both COJO and TauCOR showed low specificity, with values below 12% observed for both. For SKAT testing, TauCOR has a significantly higher specificity compared to COJO. In terms of accuracy, TauCOR showed an advantage over COJO in all scenarios (Appendix A) (see Figure 2).

The spread of the *log*_10_(*p*) values obtained from the conditional analysis was much higher for COJO than for TauCOR (Appendix A). For example, for PCA, the standard deviation of the differences between the *log*_10_(*p*) values obtained in the initial and conditional GBA analyses varied from 11.36 to 21.01 across all scenarios for COJO and from 1.23 to 2.50 for TauCOR (for details see Appendix A).

### 3.2. Real Data Analysis Using ‘Gold Standard’ List of Genes

We performed initial GBA analysis and selected only significantly associated SNP-sets with a *p*-value < 2.5 × 10^−6^ in each gene. Among 28 GS genes, 15 were significantly associated, while among 423 genes from their surroundings 54 were significantly associated. Further conditional GBA analysis was performed for these genes. The complete GBA results are presented in Appendix A and are summarized in Table 1.

As can be seen, the sensitivity of the new method for GS genes is higher than that of COJO. The specificity of the new method is substantially higher than that of COJO. TauCOR gave false positive results in only five cases, while COJO gave false positive results three times more often.

## 4. Discussion

We introduced a new method for conditional gene-based association analysis using summary statistics, named TauCOR. Compared to COJO, the new method showed equal or higher sensitivity and specificity in the majority of the simulation experiment scenarios. For real data, TauCOR outperformed COJO in all performance measures, especially in method specificity. TauCOR is a universal method for conditional gene-based analysis because the corrected distribution of z statistics can be further used for any gene-based association test that utilizes multiple linear regression models. This allows us to conclude that TauCOR is a good alternative to the more popular COJO method.

The main idea of our method is that the objects of the correction are not SNP-level z statistics, as in COJO, but the distribution of all z statistics within a gene. Most conditional analysis methods assume that the cause of the induced association signal is the effect of several independent top variants around the gene. We assume that the induced association signal is explained by the entire region surrounding the gene. This assumption is based on the notion of a regional polygenic background, which was defined via the LD score in LD score regression [23]. Previously, the influence of the regional polygenic background on trait variability was investigated at a single SNP level. In contrast, our method controls the influence of the regional polygenic background at the gene level, i.e., simultaneously for all SNPs in a gene. We demonstrated that our method is an extension of the LD score (LDSC) regression method proposed in [23] (see Appendix A).

The new method is based on the variance component approach, which assumes the random effects of the extragenic SNPs. We extracted a component of intragenic z statistic variance explained by SNPs outside the gene to correct the LD matrix for SNPs within the gene. The derived Formula (8) for the parameters of the conditional distribution of the z statistics of intragenic SNPs can also be obtained from the formula proposed by [24], where the effects of the extragenic SNPs were assumed to be fixed (see Appendix A).

We also introduced here a new method for the direct simulation of z statistics in a gene and its surrounding region without phenotype simulation. This method uses real LD matrices for SNPs in the gene and surrounding region and three predefined parameters: *h*^2^, the number of causal SNPs, and a fraction of these SNPs in the gene. In independent studies which do not consider conditional analysis, it has been empirically shown that the direct simulation of summary statistics produces very similar results to simulation of individual data across a range of scenarios, with a substantial speedup even for modest sample sizes [25,26]. We analytically confirmed the equivalence of the distributions of z statistics directly simulated and calculated using simulated phenotypes.

The proposed TauCOR method can be applied to all genes on a genome-wide scale, not just those containing significant SNPs as required by COJO. This expands the possibility of including all genes in the gene set analysis. By correcting for the polygenic background in gene set analysis approaches that use GBA results such as MAGMA [27], TauCOR may lead to more robust gene set enrichment results.

## 5. Conclusions

A new method for conditional gene-based association analysis, TauCOR, showed equal or higher sensitivity, specificity, and accuracy in the analysis of simulated and real data compared to COJO. The TauCOR method may become a good alternative to the more popular COJO method.

## Figures and Tables

**Figure 1 genes-15-01174-f001:**
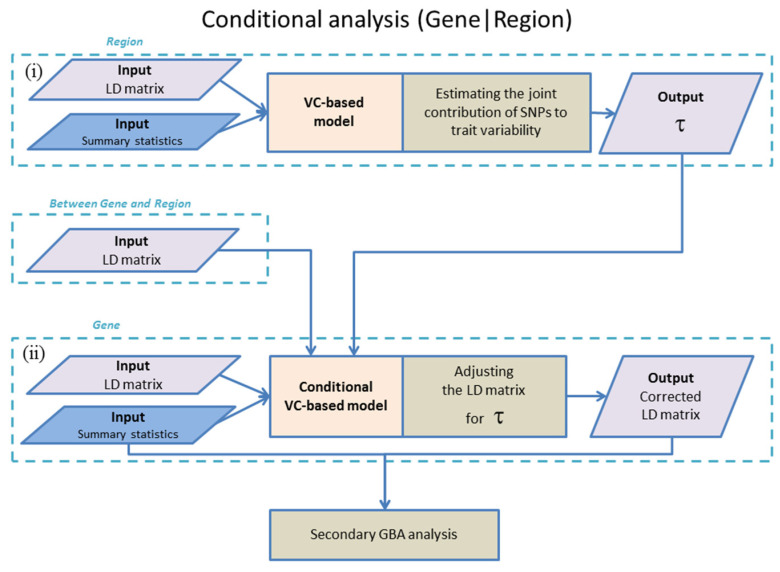
The flowchart of conditional analysis with TauCOR comprising two steps (i) and (ii).

**Figure 2 genes-15-01174-f002:**
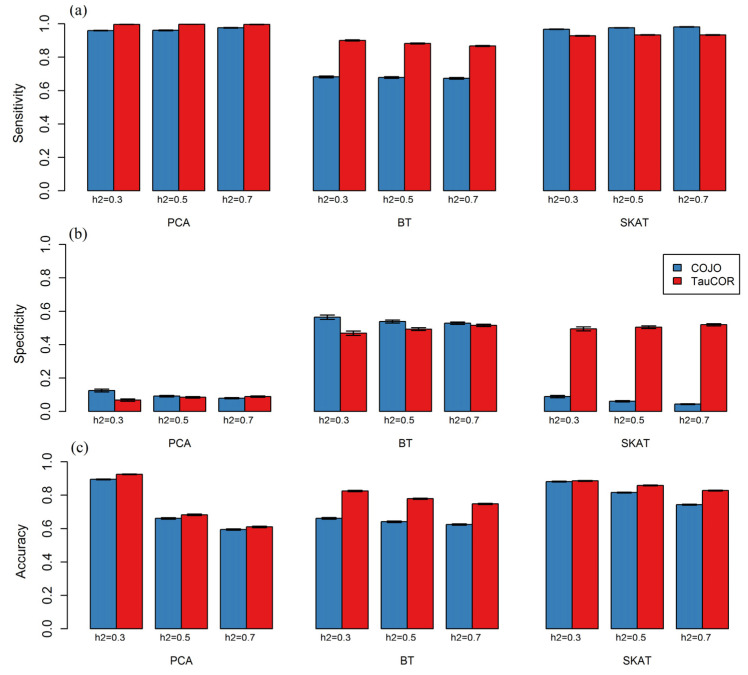
Three performance measures for the COJO and TauCOR methods of conditional GBA analysis, based on the three values of heritability. (**a**) Sensitivity calculated across the scenarios when *ρ* = 1. (**b**) Specificity calculated across the scenarios when *ρ* = 0. (**c**) Accuracy calculated as the linear combination of sensitivity and specificity.

**Table 1 genes-15-01174-t001:** Effectiveness indicators of two conditional gene-based analysis methods.

	Initial GBA	COJO + GBA	TauCOR + GBA
GS genes	15/28 *	10/15	12/15
Neighboring genes	54/423	15/54	5/54
Sensitivity	0.54	0.67	0.80
Specificity	0.87	0.72	0.91
Accuracy	0.85	0.71	0.88

*** A fractional dash is employed to separate the two numbers, indicating the number of genes that have passed a certain significance threshold and the total number of genes included in the analysis.

## Data Availability

The data are presented in the manuscript and Appendix A. The scripts implementing TauCOR and the scripts for simulating the summary statistics are posted on GitHub https://github.com/gulsvi/TauCOR (accessed on 22 July 2024).

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
