# Peer review of "A New Method for Conditional Gene-Based Analysis Effectively Accounts for the Regional Polygenic Background"

_genes, 2024, doi:10.3390/genes15091174_

Round 1

Reviewer 1 Report

Comments and Suggestions for Authors

In this paper, the authors proposed a new approach to conduct conditional gene-based analysis for testing the association between the intragenic SNPs and phenotypes while eliminating the impact of extragenic SNPs. I have the following comments below:

1. In Section 2.1.2., It is unclear to me why the effects of SNPs from the region around the gene of interest (setR) are modeled by random effects only. Could the authors explain why this assumption makes sense, particularly in the biological sense? 

2. In my understanding, colinearity, as the authors claimed to be one of the major drawbacks of COJO, is a fundamental identification issue for any statistical model that works on highly correlated features/predictors. Could the authors explain why their approach can avoid the problem with a simple linear mixed model?

Author Response

In the attached file, we provide a point-by-point response to the reviewers’ comments and concerns. In normal text: the reviewer’s comments. In italic text: taken actions.

Reviewer 2 Report

Comments and Suggestions for Authors

This paper proposes a new TauCOR method for conditional gene-based analysis using summary statistics. The idea is to correct the whole linkage disequilibrium matrix attributed to the gene rather than correcting the summary statistics for each SNP individually, as is done in the GCTA-COJO method. Simulation studies and a real-data application were conducted to evaluate the performance of the proposed method. Overall, the paper is well-motivated. My comments are listed below:

1.     Could the authors explain why they treat the effect of SNPs from the region around the gene Gr as random rather than fixed?

2.     On page 4, line 134, the authors mention, “Model (1) can be reformulated in terms of summary-level data (for details, see Appendix A).” I checked the supplementary material but could not find the details. Could the authors verify if this information is missing? The reformulation from equation (1) to (2) is not straightforward, and providing additional explanations would help readers better understand the methods.

3.    Is τ missed in expression (4), line 151 on page 4? Should it be E(z_r z_r^T )=τU_r U_r+U_r? 

4.    On line 188 of page 5, the authors mention that ρ is an indicator of the location of causal SNPs, which seems incorrect. In fact, ρ presents the proportion of causal SNPs in the gene. Could the authors double-check this?

5.     Could the authors explain why the GBA significance threshold they use is 2.5 * 10^{-6}rather than the commonly used 5 * 10^{-8}?

6.    The simulation shows results for  ρ=0 and  ρ=1, Have the authors considered testing other proportions, such as 0.5 or 0.8? It would be interesting to see how TauCOR performs with different causal SNP proportions within the gene.

Author Response

(The authors gave the same response as above.)

Round 2

Reviewer 1 Report

Comments and Suggestions for Authors

I have no concerns.

Reviewer 2 Report

Comments and Suggestions for Authors

The authors have kindly addressed all my questions, and I have no further inquiries. Thank you.